# Sympathetic Denervation Alters the Inflammatory Response of Resident Muscularis Macrophages upon Surgical Trauma and Ameliorates Postoperative Ileus in Mice

**DOI:** 10.3390/ijms22136872

**Published:** 2021-06-26

**Authors:** Shilpashree Mallesh, Reiner Schneider, Bianca Schneiker, Mariola Lysson, Patrik Efferz, Eugene Lin, Wouter J de Jonge, Sven Wehner

**Affiliations:** 1Department of Surgery, University Hospital of Bonn, 53105 Bonn, Germany; shilpashree.mallesh@ukbonn.de (S.M.); reiner.schneider@ukbonn.de (R.S.); bianca.schneiker@ukbonn.de (B.S.); mariola.lysson@ukbonn.de (M.L.); patrik.efferz@ukbonn.de (P.E.); wouter.de_jonge@ukbonn.de (W.J.d.J.); 2Department of Biology, Johns Hopkins, Krieger School of Arts and Sciences, Baltimore, MD 3400, USA; eugene.e.lin@gmail.com; 3Tytgat Institute of Liver and Intestinal Research, Academic Medical Center, 1105 Amsterdam, The Netherlands

**Keywords:** sympathetic denervation, CX3CR1^+^ macrophages, muscularis externa, neuroimmune interactions, postoperative ileus

## Abstract

Interactions between the peripheral nervous system and resident macrophages (MMs) modulate intestinal homeostatic functions. Activation of β2-adrenergic receptors on MMs has been shown to reduce bacterial challenges. These MMs are also crucial for the development of bowel inflammation in postoperative ileus (POI), an iatrogenic, noninfectious inflammation-based motility disorder. However, the role of the sympathetic nervous system (SNS) in the immune modulation of these MMs during POI or other noninfectious diseases is largely unknown. By employing 6-OHDA-induced denervation, we investigated the changes in the muscularis externa by RNA-seq, quantitative PCR, and flow cytometry. Further, we performed transcriptional phenotyping of sorted CX3CR1^+^ MMs and *ex vivo* LPS/M-CSF stimulation on these MMs. By combining denervation with a mouse POI model, we explored distinct changes on CX3CR1^+^ MMs as well as in the muscularis externa and their functional outcome during POI. Our results identify SNS as an important mediator in noninfectious postoperative inflammation. Upon denervation, MMs anti-inflammatory genes were reduced, and the muscularis externa profile is shaped toward a proinflammatory status. Further, denervation reduced MMs anti-inflammatory genes also in the early phase of POI. Finally, reduced leukocyte infiltration into the muscularis led to a quicker recovery of bowel motility in the late phase of POI.

## 1. Introduction

For the past two decades, increasing knowledge in understanding the immune modulation of the peripheral autonomic nervous system was acquired defining the enteric (ENS), sympathetic (SNS), and parasympathetic nervous system (PNS) as the three anatomically distinct branches of the autonomic nervous system. In the late 1990s, great progress has been made in studying the immunomodulatory functions of the peripheral nervous system, particularly via the vagus nerve in several inflammatory diseases including the immune-driven disorders of the GI tract [1,2,3]. Further studies in sepsis showed that the vagus nerve, as well as the SNS, controls excessive inflammation [4,5]. However, the underlying mechanism of how the SNS affects different immune-driven diseases is still not fully understood and is part of the current investigations.

Recent studies provided evidence for a regulatory role of the SNS in immune cell activation within the mucosal lamina propria [6] and in the muscularis externa wherein the SNS influences resident CX3CR1^+^ macrophage (MMs) tissue-protective programs via β2-adrenergic receptors (*Adrb2*) during bacterial infections [7]. These MMs form a dense network in proximity to the enteric and sympathetic nerves and differ from lamina propria macrophages [8,9]. The lamina propria macrophages express pro-inflammatory genes *IL-1β* and *IL-12b*, while the MMs exhibit wound healing genes such as *CD163, Retnla*, and *IL-10* [7]. Recently, it was shown that MMs upregulate neuroprotective programs upon infection via *Adrb2* through *Arg1*-polyamine secretion counteracting the loss of excitatory enteric neurons [10]. Members of our group recently also demonstrated a direct immunosuppressive effect of sympathetic neurotransmitters to microbial stimuli in murine macrophages *in vitro*, and sympathectomized mice showed spontaneous signs of colitis *in vivo* [6]. While the role of adrenergic signaling on the differentiation stage has been elegantly described in infectious settings, the role of sympathetic neurons on the MMs activation state in noninfectious, acute intestinal inflammations has not been studied so far.

Acute, noninfectious inflammation of the GI tract frequently occurs in response to intestinal manipulation (IM). The surgical handling of the bowel has been shown to activate MMs, which in turn trigger and maintain an acute inflammation leading to postoperative ileus (POI) [11], a transient motility disorder of the muscularis externa [12,13]. In the early phase of POI, MMs [11,14,15] and enteric glia cells [16,17] are activated and mediate the release of proinflammatory cytokines. The late phase is characterized by extravasation of blood-derived leukocytes, more precisely characterized as CD45^+^ Ly6C^+^ Ly6G^-^ monocytes and CD45^+^ Ly6C^+^ Ly6G^+^ neutrophils [18,19]. Both the resident and infiltrated immune cells determine the postoperative immune response and severity of POI [13] and, infiltrating monocyte-derived macrophages are shown to acquire an anti-inflammatory phenotype [19] similar to MMs under homeostasis. Although the SNS has been known for a long time as an inhibitory component in the postoperative recovery of bowel motility, potential immune-modulatory roles on the resident and infiltrating cells in this setting have not been analyzed.

In the current *in vivo* study, we aimed to provide functional insight on the role of the SNS in regulating MMs immune function during health and disease. We first compared three different sympathectomy (STX) models and identified a chemical approach as the most appropriate one. Further, we studied the effect of the latter on the immune response in the muscularis externa MMs during POI. We show that the SNS regulates MMs anti-inflammatory state in the noninflamed as well as in the acute inflammatory *conditions in vivo* and that a chemical STX has a beneficiary role in improving the late-phase inflammation during POI. 

## 2. Results

### 2.1. Chemical Depletion by 6-Hydroxydopamine (6-OHDA) Specifically Targets TH^+^ Neurons without Affecting the Vagal Innervation

In the first experimental series, we validated the efficacy of three different STX procedures by TH immunofluorescence in muscularis externa whole mounts. We compared a genetic (gSTX), a surgical (sSTX), or a chemical (cSTX) approach to deplete TH^+^ sympathetic neurons. For gSTX, we used *TH-Cre;TrkA ^f/f^* mutant mice as *TrkA* is required for the survival of sympathetic neurons, and reduction of TH^+^ cell fibers in the intestine has been microscopically observed in immunohistochemical TH stainings [20]. However, in our study, we did not find any difference in the TH expression in ileal muscularis externa whole mounts between the control *TrkA ^f/f^* and *TH-Cre;TrkA ^f/f^* mutant mice (Appendix A). In the sSTX approach, we surgically destroyed the sympathetic innervation along the mesentery artery. Two weeks after the sSTX procedure, we detected an effective ablation of TH expressing nerve fibers compared to sham-operated mice (Figure 1A). In the cSTX approach, we injected the neurotoxin 6-OHDA intraperitoneally. Two weeks later, we again observed a substantial loss of TH^+^ sympathetic fibers in whole-mount preparations of the small bowel (Figure 1B). Consistently, sections of the terminal ileum, colon, and stomach (Appendix A) also showed a loss of TH^+^ neurons, and *TH* mRNA levels were decreased in small-intestinal muscularis, lamina propria mucosa, and colon muscularis (Figure 1C–E). In addition, small bowel TH protein levels were also reduced (Appendix A). Together, both sSTX and cSTX were proven to be efficient in reducing sympathetic innervation in the GI tract.

Along the superior mesenteric ganglion, parasympathetic vagal fibers were detected [21], and the vagus nerve is also well known to exert profound immune modulatory effects [22]. Therefore, we tested if our sSTX or cSTX procedures not only target sympathetic but also vagal nerve fibers. Since there are no exclusive histological markers for the vagus nerve, we performed the functional cholecystokinin (CCK) test. CCK is a peptide hormone that has a stimulatory effect on the vagus nerve and is known to induce satiety when administered intraperitoneally to mice [23]. Therefore, mice with intact vagus nerve signaling show a reduced food consumption after CCK injection, while mice with disturbed vagus signaling should eat significantly more food. The experimental setting for the CCK test is shown in Figure 1G. Both vehicle- and cSTX-treated mice showed a reduced (−40%) food intake after CCK administration (Figure 1H), indicating that vagal signaling is still intact. In the surgical denervation experiment, sham-operated mice injected with CCK also consumed less food (−70%) (Figure 1I). However, mice that underwent sSTX did not stop eating upon CCK injection and even expressed an increased food intake (+50%) (Figure 1I). Together, these data indicate that sSTX simultaneously disturbs the vagal innervation, which is an important caveat of sSTX. By contrast, cSTX only affected the sympathetic but not the vagal innervation. Consequently, we used the cSTX to deplete sympathetic innervation in all subsequent experiments.

### 2.2. cSTX Induces a Transient Infiltration of Leukocytes into the Muscularis Externa

As 6-OHDA induces cell death of sympathetic neurons, we checked if this treatment results in an inflammation of the muscularis externa for the clearance of dying neurons. We subjected mice to cSTX and analyzed infiltrating CD45^+^ leukocytes at multiple time points by flow cytometry (Figure 2A). Subpopulations of CD45^+^ cells were identified by Ly6C^+^, F4/80^+^, and Ly6G^+^ staining. The gating strategy is shown in Figure 2B and is based on singlets and living CD45^+^ cells. After 6-OHDA treatment, the total number of CD45^+^ leukocytes in the muscularis were significantly higher at day 4 and 7 (3-fold and 4-fold, respectively, Figure 2C) indicating that immune cells infiltrate into the muscularis. On day 4, we observed significantly higher levels of Ly6C^+^ Ly6G^-^ monocytes (12-fold, Figure 2D–F) Ly6C^+^ Ly6G^+^ neutrophils (3.5-fold, Figure 2D,E,G), and F4/80^+^ macrophages (5-fold, Figure 2H). On day 7, F4/80^+^ cells (8-fold) were still elevated, while levels of monocytes and neutrophils did not increase anymore. At days 10 and 17 upon 6-OHDA treatment, all leukocyte populations returned to the level of control animals. Subsumed, intraperitoneal 6-OHDA treatment sufficiently eliminates TH^+^ fibers projecting into the small bowel muscularis externa. While the accompanying self-limiting muscularis inflammation quickly resolves, the absence of TH^+^ neurons is long-lasting. To avoid any cellular side effects due to this inflammation, we waited 17 days after cSTX for any further analyses.

### 2.3. Sympathetic Denervation Alters the Inflammatory State of CX3CR1^+^ MMs in a Noninfectious Setting

By bulk 3′ mRNA sequencing, we next analyzed the molecular profile within the muscularis externa upon cSTX. This analysis was performed 17 days after 6-OHDA treatment when the transient leukocyte muscularis infiltration was cleared for at least 7 days. Principal component analysis showed that the vehicle- and cSTX-treated samples clustered into two different groups (Figure 3A), indicating significant changes induced by cSTX. Heat map analysis of differentially expressed genes showed the hierarchical clustering between two groups (Figure 3B). In total, 1740 genes were differentially expressed, of which 775 were upregulated and 965 were downregulated in the cSTX-treated group as compared to vehicle-treated (Figure 3C). Gene enrichment analysis identified the immune-related gene ontology terms “defense response, inflammatory response, immune response, positive regulation of inflammatory response and response to cytokines” to be enriched upon 6-OHDA-treatment (Figure 3D), suggesting an altered immune state of the muscularis upon sympathetic neuronal loss. By using qPCR analysis, we confirmed the altered immune state with a gene expression panel of pro- and anti-inflammatory genes that we recently identified to be regulated during acute inflammation of the muscularis [19]. Higher expression of proinflammatory genes such as *IL6, IL-1β*, and *TNF* was detected in cSTX-treated mice (Figure 3E). For the anti-inflammatory gene panel, *CD163* and *YM1* showed comparable levels between vehicle- and cSTX-treated mice, while *Retnla1* was down and *Arg1* was upregulated. As adrenergic receptors are stimulated by norepinephrine, we checked if sympathetic denervation had any effect on these receptors. We observed altered adrenergic receptor levels in the muscularis (Figure 3F) and mucosa (Figure 3G) after cSTX. Most adrenergic receptors were downregulated in the mucosa, while in the muscularis externa some were down and others were upregulated. Taken together, these data indicate that sympathetic neuronal loss shapes the basal immunological gene expression toward a proinflammatory and reduced anti-inflammatory pattern.

We next hypothesized that the altered basal immune pattern upon cSTX might be due to an altered MMs activation state. Therefore, we sorted CX3CR1^+^ resident MMs from *CX3CR1 ^GFP/+^* reporter mice using saline or 6-OHDA treatment (Figure 3H). These sorted macrophages showed lower levels (−80%) of *Adrb2* (β2-adrenergic receptor) and higher levels (1.5-fold) of *Adrb3* (β3 adrenergic receptor). Further, the levels of *CSF-1R* (−40% macrophage colony receptor-1) and *BMP2* (−50%, bone morphogenetic protein-2) (Figure 3I) were reduced after cSTX, indicating that the macrophage to neuron communication [8] might also be disturbed by sympathetic loss. Notably, on this individual cellular level, we also observed a reduction in anti-inflammatory genes *CD163* (−75%), *YM1* (−50%), and *Retnla1* (−60%) after cSTX, while proinflammatory genes *IL6, IL-1β, TNF*, and *iNOS* did not change (Figure 3J).

Based on these data, we speculated that sympathetic neuronal loss dampens the anti-inflammatory capacity of intestinal CX3CR1^+^ MMs. This hypothesis was tested by using an *ex vivo* setup, wherein we sorted and stimulated CX3CR1^+^ MMs from saline or cSTX-treated mice with LPS or M-CSF (Figure 3K). This experiment should show alterations in the MMs anti-inflammatory or pro-inflammatory responses induced by M-CSF or LPS, respectively. Comparable expression of *IL-1β* and *TNF* and a slightly increased *IL6* expression were observed in both groups in macrophages after LPS stimulation (Figure 3L). By contrast, the anti-inflammatory genes *Arg1, CD163*, and *IL10* were reduced in cSTX-sorted M-CSF-treated macrophages. This is in line with the cSTX-induced reduction of *CSF-1R* receptor in sorted CX3CR1^+^ MMs (Figure 3I). Together, our findings show that the depletion of sympathetic neurons results in a pro-inflammatory signature on the intestinal CX3CR1^+^ macrophages, and their anti-inflammatory genes are reduced.

### 2.4. Sympathetic Denervation Reduced CX3CR1^+^ Muscularis Macrophage Anti-Inflammatory Response in an Acute Intestinal Postoperative Ileus Model

After having shown that the SNS affects transcriptional programs of MMs toward pro-inflammatory signature within the muscularis externa, we next asked if it has any consequences in an acute, intestinal, noninfectious inflammation. To answer this, we took advantage of the well-described surgical intestinal manipulation (IM) model that functionally results in a clinically relevant motility disorder, known as postoperative ileus (POI) (Figure 4A). POI is a common consequence of abdominal surgery and is due to a macrophage-triggered acute muscularis externa inflammation leading to transient disturbances of GI motility [11]. Animals underwent 6-OHDA or vehicle treatment 17 days before the surgical IM or a laparotomy (sham-operation) (Figure 4B). After 3 h, prototypical cytokines including *IL-1β* and *IL6* were strongly increased in the muscularis by IM compared to the sham-operated groups. However, there were no differences in expression between cSTX- and vehicle-treated groups after IM (Figure 4C). In contrast, the anti-inflammatory gene *YM1* did increase (50-fold) after IM in vehicle-treated mice, but this increase was reduced (−60%) in the cSTX group (Figure 4D). *CD163* and *Retnla1* decreased after IM but no difference was observed between vehicle and cSTX. Arg1 expression declined in vehicle-treated mice upon IM (−50%), while this was not observed in the cSTX group. These findings suggest that within the muscularis, distinct immune changes could be observed after cSTX, but the effect of MMs polarization remained unclear. Therefore, we sorted intestinal CX3CR1^+^ cells 3 h after IM from vehicle- and cSTX-treated mice (Figure 4E). Three hours after IM, *IL6* decreased in sorted macrophages in both groups while *IL-1β, TNF*, and *iNOS* did not change, and no difference was observed between cells isolated from cSTX- or vehicle-treated animals (Figure 4F). In contrast, the postoperative anti-inflammatory gene levels of *Arg1* (−60%), *CD163* (−30%), and *YM1* (−60%) were significantly reduced in macrophages from cSTX animals with the only exception in *Retnla1* levels showing no alterations (Figure 4G). Taken together, these findings show that a preoperative cSTX reduces the postoperative expression of several genes associated with an anti-inflammatory MMs response. We speculated that the reduced anti-inflammatory immune status might affect the postoperative muscularis inflammation and subsequently the functional outcome of POI.

### 2.5. Sympathetic Denervation Reduced the Infiltration of Immune Cells into the Muscularis Externa and Led to an Accelerated GI Transit Time

To analyze the role of sympathetic innervation in POI, we measured the outcome of mice subjected to cSTX or vehicle in the clinically relevant effector phase of POI. This phase manifests around 24 h after IM in the POI mouse model when cellular inflammation and functional disturbances peak (Figure 5A). Through 3′ bulk mRNA sequencing of the muscularis, we identified 456 differentially expressed genes between cSTX and vehicle-treated mice 24 h after IM, and more than half of them increased in the cSTX mice (Figure 5B,C). GO terms for “antimicrobial humoral response, inflammatory response, defense response, acute inflammatory response, macrophage polarization and positive regulation of inflammatory response” were also enriched in cSTX group (Figure 5D), showing that cSTX alters the immune state of the muscularis upon IM also in the effector phase of POI. Individual gene expression analyses by qPCR revealed upregulation of *IL-1β* and *IL6* 24 h after IM in vehicle-treated mice (Figure 5E). Both genes were further increased *IL-1β* (2.5-fold), *IL6* (3.5-fold) in the cSTX group. Simultaneously, the anti-inflammatory genes *CD163* (−65%) and *YM1* (−30%) decreased, while no changes occurred in *Arg1* and *Retnla1* expression 24 h after IM (Figure 5F). This shows that the effects of sympathetic denervation on the postoperative immune response last for hours and might also affect the functional postoperative outcome of POI. To analyze the cellular and functional outcome of cSTX in POI, we again subjected mice to cSTX or vehicle treatment before IM or sham operation and measured the cellular infiltration and motility disturbances. The latter was investigated by measurement of the gastrointestinal (GI) transit time. Twenty-four hours after IM, vehicle-treated animals showed a reduced GI transit (3.3 ± 0.26) compared to sham-operated animals (10.8 ± 0.45). In contrast, cSTX-treated animals showed an accelerated GI transit (5.7 ± 0.75) compared to the vehicle-treated mice 24 h after IM (Figure 5G). Notably, leukocyte numbers correlated inversely with a reduction (−55%) in postoperative MPO^+^ immune cell infiltration into the muscularis of cSTX mice 24 h after IM (Figure 5H,I).

Further characterization of the postoperative muscularis immune cells (Figure 6A) showed a reduction of living Ly6C^+^ Ly6G^-^ monocytes, Ly6G^+^ Ly6C^+^ neutrophils (Figure 6B) and monocyte-derived F4/80^+^ macrophages (Figure 6C) compared to vehicle-treated animals. Quantification revealed a 37% reduction in living CD45^+^ leukocytes, 34% reduction in Ly6C^+^ Ly6G^-^ monocytes, 46% reduction in Ly6G^+^ Ly6C^+^ neutrophils, and 41% reduction in monocyte-derived F4/80^+^ macrophages (Figure 6D–G) in cSTX-treated compared to vehicle-treated animals 24 h after IM. Together, the depletion of sympathetic neurons resulted in a reduced number of infiltrating leukocytes after IM and consequently led to an improved GI transit.

In conclusion, this study demonstrates that the sympathetic nervous system distinctly shapes the immune response of MMs under homeostasis and inflammation. Depletion of the sympathetic innervation affects the MMs immune state and thus reduces cellular inflammation leading to an improved postoperative outcome of POI.

## 3. Discussion

The SNS is well known to regulate the motility of the GI tract together with the parasympathetic nervous system. Sympathetic overactivity occurs upon abdominal surgery disturbing the motility patterns resulting in POI [24]. Recent studies highlighted a specific role of the SNS in modulating MMs immunological function during Salmonella infection [7,10]. As we and others have previously shown that MMs orchestrate the acute postoperative, noninfectious inflammation during POI [11,19,25], we hypothesized that the SNS controls the MMs immune status during surgery-induced POI. Accordingly, in this study, we focused on understanding the immunomodulatory role of the SNS and its interaction with MMs under homeostasis and during POI *in vivo*.

Several techniques have been used to eliminate the sympathetic innervation *in vivo*. To find the most appropriate model for intestinal sympathetic denervation, we employed a genetic, surgical, and chemical sympathetic ablation to accomplish effective STX in the bowel wall and particularly within the intestinal muscularis externa. A genetic approach would have the advantage of not requiring any chemical or surgical intervention that might affect other analyses. The genetic approach tested herein was effective in reducing sympathetic neurons in the pancreas via deficiency of the nerve growth factor receptor *TrkA* in TH^+^ sympathetic neurons [20] and further to a not quantified amount also in the murine intestine. However, we did not detect a reduction of TH^+^ fibers in the muscularis externa of the *TH-Cre;TrkA ^f/f^* mutant mice. Due to the successful use of TH-Cre mouse lines in GI research [26], we speculate that *TrkA* deficiency might not be required for the survival of sympathetic nerves innervating the GI tract or that compensatory mechanism to *TrkA* action might exist in the intestine compared to the pancreas. The alternatively tested surgical approach [27] is selective and site-specific, although surgical transection is at risk of also disturbing the fine bundles of vagal afferent fibers that are dividing and reassembling along the superior mesenteric artery [21]. While vagal fibers are hardly detectable at the quantitative levels due to the missing vagal markers, we used an indirect functional CCK response test to determine vagal integrity by assessing food intake. An intact vagus nerve mediates satiety upon CCK administration, and this effect is blocked by vagotomy [28]. As CCK did not induce satiety upon surgically sympathectomized mice, we concluded that vagal signaling was disturbed by the sSTX surgery. Finally, cSTX, based on the neurotoxic action of 6-OHDA, turned out to be the most suitable approach. This approach is widely used to selectively damage and deplete catecholaminergic neurons [29]. The advantage in our study was the almost complete loss of TH^+^ fibers in the muscularis externa at different sites of the GI tract (only a few TH^+^ fibers remaining) and the absence of any effects on CCK-mediated satiety induction. Although the CCK test is only an indirect measure of vagus nerve integrity, other reasons might be responsible for the unresponsiveness to CCK stimulation in the sSTX approach. Nevertheless, vagus nerve signaling was not disturbed by the 6-OHDA treatment. A disadvantage of the 6-OHDA model is that it also targets both dopaminergic and noradrenergic neurons [30] and might also act at distant sites due to the peritoneal application route. Nevertheless, in the muscularis externa, we found a transient macrophage and monocyte increase within the first 4–7 days after 6-OHDA treatment. Neutrophil numbers were not elevated. We see this as a sterile inflammatory response that might be required for the clearance of cellular debris from dying sympathetic neurons. However, the cellular inflammation was transient in nature and disappeared after 10 days, a whole week before we started any of our analyses. Thus, cSTX appears to be a suitable technique to eliminate the sympathetic innervation in the GI tract and potentially also in other visceral organs without disturbing the vagal innervation. However, researchers should individually examine their mouse model for the presence and duration of a 6-OHDA-induced inflammation to avoid any side effects, particularly in immunological analyses. Notably, we cannot exclude any long-term effects of the cSTX-induced initial inflammation, which might desensitize the bowel wall to another immune stimulus even 10 days after the initial inflammation was completely resolved.

By subjecting *CX3CR1^GFP/+^* reporter mice to the cSTX procedure, we revealed an altered response of CX3CR1^+^ MMs receptor and cytokine expression. Notably, cSTX strongly affected gene expression levels for distinct adrenergic receptors (AR) in the muscularis externa tissue but also in sorted macrophages. Two ARs that are expressed by various immune and nonimmune cells in the GI tract [31] and regulate intestinal inflammation [32,33] are *Adrb2* (β2AR) and *Adrb3* (β3AR). These ARs were upregulated in the muscularis externa but strongly downregulated in the sorted MMs, indicating a compensatory response of other muscularis cells. The reduction of Adrb2 in MMs upon cSTX is in line with previous studies identifying the *Adrb2* directing MMs inflammatory state upon bacterial infection [7]. Simultaneous to the decrease in *Adrb2*, the *Adrb3* was slightly upregulated in MMs upon cSTX. It remains unclear if this is part of a compensatory mechanism under catecholaminergic deficiency. Notably, *CSF1-R* and *BMP2* were reduced after cSTX. *CSF-1R* is essential for MMs maintenance, and BMP-2 is a soluble growth factor secreted by MMs which regulates the development of enteric neurons and GI motility in a noninflamed bowel [8] Interestingly, the CX3CR1^+^ MMs, stimulated *ex vivo* with M-CSF, express much lower levels of the alternative activation markers *IL-10, Arg1*, and *CD163*. Previous studies showed that stimulation with CSF-1 polarizes macrophages away from an antigen-presenting phenotype and toward an immunosuppressive function in both mice and humans [34]. Therefore, the altered *CSF-1R* expression might be interpreted as a part of the mechanism behind the reduced anti-inflammatory capacity of MMs in the absence of functional sympathetic innervation. However, as CSF-1 signaling is rather complex under inflammatory conditions [35], further studies are required to make a stronger point on the CSF-1 effects, particularly *in vivo*. The reduced *BMP2* levels might explain the slightly, but not significantly, reduced GI motility under baseline conditions as observed in our study as well as others studying *Adrb2* deficiency [7].

Besides these changes, the baseline macrophage anti-inflammatory phenotype was also altered by cSTX, as observed by a reduced expression of markers that are commonly used to identify an alternatively activated macrophage state: including *YM1, CD163*, and *Retnla1* [36]. Although these markers were predominantly reduced in the sorted MMs, this reduction was not detected in the complete muscularis externa tissue. Conversely, cSTX led to an upregulation of the proinflammatory mediators *IL-6* and *IL-1β* in the muscularis externa, although this was not confirmed in the sorted macrophages. This indicates that the SNS triggers a pro-inflammatory response in the muscularis externa where these MMs reside. *Arg1* was upregulated in peritoneal macrophages following NE or salbutamol stimulation *in vitro*, while the presence of *β2AR* blocker butaxamine prevented this increase [7]. However, *Arg1* gene expression was not altered *in vivo* by cSTX. Nevertheless, many other prototypical anti-inflammatory markers, which have not been tested in the aforementioned *in vitro* study, were reduced by cSTX, and this supports the generally proinflammatory role of the SNS in MMs *in vivo*. This was finally supported by our *ex vivo* LPS or M-CSF stimulation study. Both molecules were previously shown to induce a pro- or anti-inflammatory cytokine profile, respectively [37,38]. Flow cytometry sorted MMs from the vehicle- and cSTX-treated mice expressed comparable levels of pro-inflammatory genes upon an *ex vivo* LPS stimulation, while MMs from cSTX-treated mice failed to respond to M-CSF. This shows that MMs remain in a proinflammatory state *in vivo* following cSTX and are resilient to anti-inflammatory stimuli such as M-CSF.

Several studies suggest that the underlying mechanism of the SNS on macrophages depends on direct adrenergic pathways, particularly via β2-adrenergic receptors on macrophages. The close proximity of MMs to TH^+^ extrinsic sympathetic neurons is a prerequisite. The observation of a NE-mediated β2-adrenergic induction of anti-inflammatory neuroprotective programs in resident macrophages during bacterial challenges further supports the hypothesis of a direct interaction [7,10]. Furthermore, butaxamine, an *Adrb2* receptor antagonist, prevented this increase in macrophage anti-inflammatory genes *in vitro* [7]. Therefore, the shift in macrophage immune status upon denervation is highly likely due to the missing adrenergic part of the sympathetic innervation. Nevertheless, sympathetic neurons release further neurotransmitters such as neuropeptide Y (NPY), somatostatin, and other neuropeptides [39] that might also affect macrophages’ immune status. Additional experiments, e.g., in mice deficient of *Adrb2* in macrophages, could finally clarify the mechanistic role of SNS on macrophage function.

While our results, together with the previous data from others, now provide evidence that the SNS affects the MMs immune phenotype under baseline conditions *in vivo*, we tried to understand its role in inflammatory conditions. A negative immune regulatory role of SNS has been shown in the murine acute peritonitis model [40], and sympathectomized mice even developed clinical signs of colitis [6]. In abdominal surgery, inhibitory sympathetic neuronal pathways become activated and are known to trigger dysmotility of the GI tract, clinically known as POI [41,42,43]. However, the mechanism behind the sympathetic activation in POI development has never been addressed in terms of immunological changes that are crucial to POI development [11,14]. We observed that mice who underwent an IM after cSTX expressed enhanced inflammatory responses. Notably, these changes were detected in the early phase 3 h after IM when MMs are activated and release pro-inflammatory cytokines [14], but tissue infiltration of blood-born leukocytes has not yet occurred at this time point [44]. Interestingly, cSTX led to a pro-inflammatory immune response yet reduced leukocyte infiltration of the smooth muscle layer and reduced POI in terms of GIT in the late phase of POI. As infiltrating leukocytes numbers declined, we anticipate that proinflammatory cytokine induction derives from resident-producing cells rather than from the infiltrating cells. For *IL-1β* and *IL6*, this could be enteric glia that is known to contribute to POI development [16,17]. Unfortunately, we were unable to measure discrete gene expression profiles from resident MMs isolated in the late phase of POI as MMs can be hardly separated from infiltrating monocyte-derived macrophages due to an overlapping marker expression. The observation that cSTX led to an increased *IL-6* and *IL-1β* gene expression but reduced the inflammatory infiltrate and improved POI is likely a reflection of the fact that IM-induced sympathetic reflexes are disturbed. A limitation of this study is that we have not been able to perform an SNS stimulation to validate whether POI would worsen if SNS stimulation is performed.

In summary, we show that the sympathetic neurons modulate MMs immune function (Figure 7). Sympathetic depletion reduced the anti-inflammatory genes of MMs and proved to be beneficial in reducing immune cell infiltration and the symptoms of an acute, noninfectious, surgically induced bowel wall inflammation of POI. Selective interventions on the sympathetic activation may serve as a possible therapeutic option for the treatment of POI or other acute immune-driven diseases.

## 4. Materials and Methods

### 4.1. Animals

In this study, 8–10 weeks old male *C57BL6/J* mice from Janvier (Saint Berthevin cedex, France) were used to perform surgical and chemical denervation and intestinal manipulation. Fixed intestinal tissue samples from *TH-Cre;TrkA^f/f^* mutant mice were generously provided by Prof. Rejji Kuruvilla (John Hopkins University, Baltimore, MD, USA). All animals were maintained under pathogen-free conditions, and experiments were carried out in accordance with federal law for animal care protection.

### 4.2. Surgical Denervation

Mice were anesthetized by inhalation of isoflurane (3–5%, 3–5 L/min flow), and 10 mg/kg tramadol was injected subcutaneously a few minutes before surgery. A midline incision of approximately 1 cm was made to open the skin and peritoneum. The entire small bowel was taken out to reach the superior mesenteric artery that is located at the posterior border of ganglia. We then carefully destroyed the sympathetic plexus that is running along the superior mesenteric artery [27]. The small bowel is placed back, and the abdomen is sutured with a 5.0 silk thread. Sham-operated mice were used as control. For sham operation, the small bowel was taken out and placed back without destroying the sympathetic plexus. Drinking water was supplied with 1 mg/kg tramadol for 3 days to both sham-operated and surgically denervated mice. Two weeks after the surgery, intestinal whole-mount immunofluorescence staining was performed to check for the absence of TH^+^ sympathetic neurons.

### 4.3. Chemical Denervation

In this study, 6-OHDA has been used for sympathetic denervation by others in various studies [6,45,46]. To achieve effective denervation, we tested different dosages of 6-OHDA including one-time 200 mg/kg, two-times 100 mg/kg, two-times 50 mg/kg, and three-times 80 mg/kg in our preliminary experiments (data not shown). We speculated that a higher dosage of 6-OHDA might be harmful to the animals; therefore, we gave lower doses of repetitive injections. Three-times 80 mg/kg 6-OHDA was the only dosage effective in depleting TH^+^ fibers. Further, these fibers were absent for at least 60 days after three-times 80 mg/kg 6-OHDA treatment. Therefore, we used this dosage for all the subsequent analyses. Mice were injected intraperitoneally with sterile 80 mg/kg 6-hydroxydopamine hydrobromide dissolved (Sigma Aldrich, Saint Louis, MO, USA) in 0.1% l-ascorbic acid-containing saline for 3 consecutive days. The vehicle group was injected with only sterile saline for 3 days. Two weeks after the last 6-OHDA injection, mice were sacrificed and checked for the absence of TH^+^ neurons by immunofluorescence staining.

### 4.4. Cholecystokinin (CCK) Test

For the CCK test, mice were starved for 20 h with only a supply of drinking water. After 20 h, 8 µg/kg CCK (Sigma Aldrich, Saint Louis, MO, USA) dissolved in NaCl was injected intraperitoneally. All the mice were caged alone and were provided with only preweighed food without the supply of drinking water. After 2 h, the amount of food consumed by each mouse was carefully estimated [47].

### 4.5. Whole-Mounts Preparation and Immunofluorescence Staining

A segment of 4 cm long distal small bowel was cut opened and fixed in 4% histofix (Carl Roth, Karlsruhe, Germany) for 20 min. The fixed tissue was washed 3 times with Krebs–Ringer buffer (KRB). Under microscopic observations (M651, Leica, Wetlar, Germany), mucosa and submucosa were stripped, and the muscularis whole mounts were used for immunofluorescence staining. Each whole mount was permeabilized with 2% Triton X-1000 (Sigma Aldrich, Saint Louis, MO, USA) for 15 min and then blocked-in donkey blocking buffer for 1 h at room temperature. The whole mounts were later incubated with primary antibodies rabbit TH (Millipore, Darmstadt, Germany) and rat MHCII overnight at 4 degrees and constant shaking. The whole mounts were washed 3 times with PBS and incubated with 1:800 dilutions of secondary antibodies donkey antirabbit cy3 (Dianova, Hamburg, Germany) and donkey antirat cy5 (CF647 Sigma, 2 mg/mL) for 1 h at room temperature. After washing the whole mounts 3 times with PBS, a few drops of Shandon Immu-mount mounting agent (Thermo Scientific, Waltham, MA, USA) was added, coverslipped (24 × 60 mm), and kept for drying in dark. Nikon Eclipse TE2000-E microscope (Melville, NY, USA) was used to take images at 20x magnification.

### 4.6. Intestinal Manipulation

Intestinal manipulation (IM) was carried out using the standardized procedure that was previously described [48]. Mice were anesthetized by inhalation of isoflurane (3–5%, 3–5 L/min flow), and 10 mg/kg tramadol was injected subcutaneously a few minutes before surgery. A midline incision of approximately 1 cm was made to open the skin and peritoneum. The entire small bowel content was taken out, placed on sterile cotton gauze, and lightly manipulated using two sterile moist cotton swabs two times from duodenum to terminal ileum. After IM, the small bowel was carefully placed back, and the abdomen was closed by two layers of continuous sutures using 5.0 silk thread. The operated mice received 1 mg/kg tramadol in drinking water until they were sacrificed.

### 4.7. Hanker Yates Staining

A segment of 4 cm long ileum was cut opened and fixed in 100% ethanol (PanReac Applichem, Darmstadt, Germany) for 10 min. The fixed segment was washed 3 times with Krebs–Ringer buffer (KRB), and mucosa and submucosa were stripped. The whole mounts were stained for 10 min with 1 mg/mL hanker yates (Polysciences, Aidenbach, Germany) dissolved in PBS buffer to detect myeloperoxidase (MPO)-positive cells. After washing them 3 times, the whole mounts are coverslipped with an aqueous mounting agent (Merck, Darmstadt, Germany) and kept for drying. MPO^+^ cells from 5 randomly chosen areas were counted at 160× magnification.

### 4.8. Gastrointestinal Transit Time Measurement

The gastrointestinal transit (GIT) was measured 24 h after the IM. Mice were gavaged with 100 µL of 6.25 mg/mL FITC-labelled dextran (70 kDa; Sigma-Aldrich, Saint Louis, MO, USA) 22.5 h after IM. Next, 90 min after the administration, mice were sacrificed, and the complete gastrointestinal tract from stomach to colon was divided into 15 segments. The luminal contents in each segment were collected separately, and the absorbance of FITC-dextran was measured. The geometric center (GC) of the distribution of FITC-labeled dextran was used to calculate the gastrointestinal transit time with the formula:$$GC=\sum \left(\%\;of\;total\;fluorescent\;signal\;per\;segment*segment\;number\right)/100$$


### 4.9. Flow Cytometry

The muscularis externa tissue from the small bowel was enzymatically digested for 40 min in a shaking water bath at 37 °C with a mixture of the following enzymes: 0.1% collagenase type II (Worthington Biochemical Corporation, Lakewood, NJ, USA) diluted in PBS containing 2.4 mg/mL dispase II (Neutral protease grade II, Roche, Mannheim, Germany), 0.1 mg/mL deoxyribonuclease I (Worthington Biochemical corporation, Lakewood, NJ, USA), 1 mg/mL bovine serum albumin (Applicher, Darmstadt, Germany), and 0.7 mg/mL trypsin inhibitor (Applicher, Darmstadt, Germany). The cell suspension was filtered with 70 µm gauze, and the single cells were stained with antimouse CD16/32 (Trustain fcX, Biolegend, San Diego, CA, USA) and 1:200 dilution of fluorochrome-labelled monoclonal antibodies against FITC antimouse CD45 (Biolegend, San Diego, CA, USA), Alexa fluor 647 antimouse Ly6G (Biolegend, San Diego, CA, USA), PE Cy7 antimouse Ly6C (Biolegend, San Diego, CA, USA), and PE antimouse F4/80 (Biolegend, San Diego, CA, USA) for 30 min at 4 °C. Hoechst-33342^+^ (Sigma Aldrich, Saint Louis, MO, USA) was used to exclude dead cells. The analysis was performed on canto I (FACS Canto, BD Biosystems, Heidelberg, Germany) at FACS core facility, Bonn, Germany, using a FACS diva software (BD Biosciences, Heidelberg, Germany), and the gating strategies are explained in the figure legends. All the data from flow cytometry were analyzed using the FlowJo software (Tree Star Inc., Ashland, OH, USA).

### 4.10. Flow Cytometry-Based Cell Sorting

The muscularis externa tissue of three CX3CR1^GFP/+^ mice was pooled together for each sort. The extraction of single cells is described above. To exclude bone marrow cells, we performed heart perfusion by flushing 30 mL of ice-cold PBS buffer (Life Technologies, Darmstadt, Germany). CX3CR1^+^ cells were acquired and sorted by FACS Aria flow cytometer (BD Biosciences, Heidelberg, Germany) at FACS core facility, Bonn, Germany.

### 4.11. Ex Vivo Stimulation Assay

CX3CR1^+^ ME macrophages were FACS^-^ sorted and cocultured with either LPS (100 ng/µL) or M-CSF (100 ng/µL) in DMEM with 10% FCS medium for 3 h at 37 °C. Cell lysis and total RNA extraction were carried out using Arcturus TM Picopure TM RNA isolation kit (Thermo Scientific, Waltham, MA, USA).

### 4.12. Western Blot

Protein lysates were prepared from the vehicle- and cSTX-treated mice muscularis externa samples. Protein concentration was estimated using standard BCA assay (Thermo Scientific, Waltham, MA, USA), and protein samples were incubated with SDS-loading buffer. Samples were analyzed by loading onto 4-20 % gradient gels (Biorad, Hercules, CA, USA) and adding a 10× NuPage reducing agent (Invitrogen, Waltham, MA, USA). By using the Transblot turbo system (Biorad, Hercules, CA, USA), the gels were transferred to PVDF membranes. The membranes were blotted in 5% skimmed milk for 1 h and then in TH polyclonal antibody (Millipore, Darmstadt, Germany) overnight at 4 °C. This is followed by incubation with horseradish peroxidase (HRP) coupled secondary antibody for 1 h at RT. The blots were imaged using the ChemiDoc imaging system (Biorad, Hercules, CA, USA) after the addition of a chemiluminescent substrate.

### 4.13. Quantitative RT-PCR

Total RNA was extracted from the muscularis tissue using trizol (Life Technologies, Darmstadt, Germany)-based approach. For flow cytometry sorted cells, total RNA was extracted using Arcturus TM Picopure TM RNA isolation kit (Thermo Scientific, Waltham, MA, USA). The extracted RNA was reverse-transcribed to cDNA using the high-capacity cDNA reverse-transcription kit (Thermo Scientific, Waltham, MA, USA). The messenger RNA (mRNA) expression was quantified in triplicates by a real-time reverse-transcriptase-polymerase chain reaction (RT-PCR) with TaqMan or SYBR green probes. The PCR reaction mixture was prepared using the TaqMan gene expression master mix (Thermo Scientific, Waltham, MA, USA) or SYBR green PCR master mix (Appendix A).

### 4.14. RNA Sequencing

The sequencing of total RNA isolated from the muscularis tissue was carried out using next-generation sequencing (NGS) at the NGS core facility, University Hospital of Bonn. RNA-seq data were subjected to quality control using FASTQC. Partek flow software (Partek, St. Louis, MO, USA) was used to generate heatmaps and volcano plots. Lexogen 12,112,017 pipeline was used for the analysis. STAR 2.5.3a aligner with Mus musculus—mm10 index was aligned to Ensembl Transcripts release 95 to generate transcript counts.

### 4.15. Statistical Analysis

The statistical analysis was performed with GraphPad Prism version 8.4.3 software (GraphPad, San Diego, CA, USA) using an unpaired *t*-test, one-way or two-way multiple comparison ANOVA test. Significance levels were indicated as *p* ≤ 0.05 (*), *p* ≤ 0.01(**), and *p* ≤ 0.001(***). Values were expressed as mean + SEM.

## Figures and Tables

**Figure 1 ijms-22-06872-f001:**
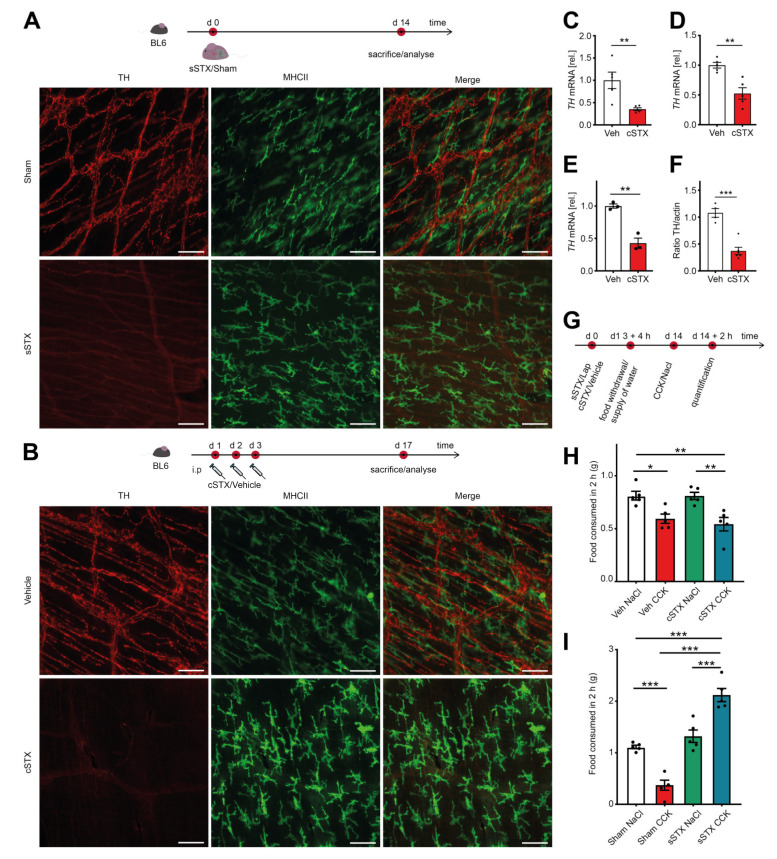
Analysis of the different sympathectomy models. (**A**) The experimental setting for the surgical (sSTX) model. Two weeks after surgical denervation at the level of the superior mesenteric artery, a significant reduction in TH^+^ fibers was observed in ileal whole-mounts as compared to the sham-operated mice. Scale bars, 100 μm. (**B**) The experimental setting for the chemical (cSTX) model. Two weeks after 6-OHDA-treatment, TH^+^ fibers were effectively depleted as shown by an absence of TH immunofluorescence in ileal whole-mounts compared to vehicle-treated mice. Scale bars, 100 μm. (**C**–**E**) qPCR analysis showing the downregulation of *TH* mRNA in small-intestinal muscularis (*n* = 5) (**C**), small-intestinal lamina propria (*n* = 5) (**D**), and colon muscularis (*n* = 3) (**E**) in cSTX-treated mice (red bar) as compared to the vehicle-treated mice (white bar). The graphs are presented relative to GAPDH house-keeping gene of vehicle-treated mice. The statistical analysis was carried out by unpaired *t*-test (* *p* < 0.05, ** *p* < 0.01, and *** *p* < 0.001) (**F**) Quantification of the TH protein showing reduced protein levels in cSTX-treated mice (red bar) compared to vehicle-treated mice (white bar). The graphs are normalized to actin protein bands and are plotted as mean ± SEM. The statistical analysis was carried out by unpaired *t*-test (* *p* < 0.05, ** *p* < 0.01, and *** *p* < 0.001) for (**C**–**F**) comparing vehicle-treated mice versus indicated groups. (**G**) The experimental setting for the CCK test. (**H**) 6-OHDA-treated mice injected with CCK showed a significant reduction in food consumption indicating intact vagus nerve signaling upon cSTX (*n* = 5). (**I**) Surgically denervated mice injected with CCK showed a significant increase in food consumption as compared to sham-operated mice (*n* = 5). Values in each column in (**H**,**I**) are shown as mean ± SEM, and statistical analysis was carried out by two-way ANOVA (* *p* < 0.05, ** *p* < 0.01, and *** *p* < 0.001).

**Figure 2 ijms-22-06872-f002:**
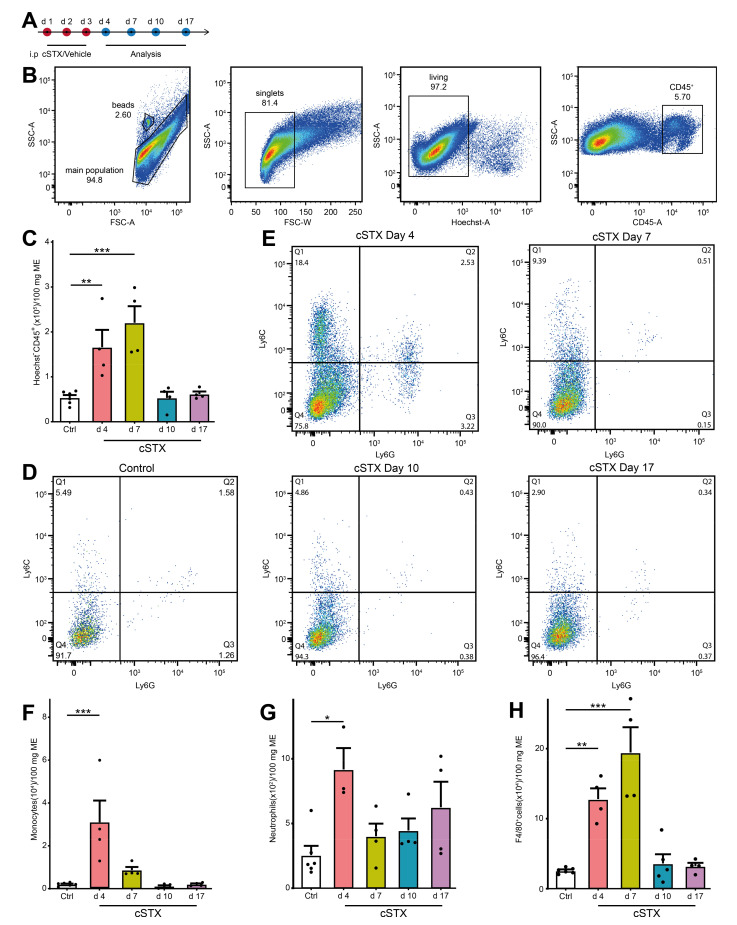
cSTX induces a transient inflammation of the muscularis externa. (**A**) The experimental setting for the analysis of infiltrating immune cell population into muscularis externa by FACS after cSTX-treatment. (**B**) Representative FACS images showing the gating strategy of the main population, singlets, and living CD45^+^ cells. (**C**) Bar graphs showing the absolute numbers of total Hoechst- CD45^+^ cells isolated from the small bowel muscularis externa after cSTX-treatment (*n* = 5, 4). (**D**,**E**) Representative FACS images showing the ratio of Ly6C^+^ Ly6G^-^ monocytes, Ly6C^+^ Ly6G^+^ neutrophils of control (**B**), and cSTX-treated mice (**C**) at day 4, 7, 10, and 17. (**F**–**H**) Bar graphs showing the absolute numbers of total Ly6C^+^ Ly6G^-^ monocytes (**F**), Ly6C^+^ Ly6G^+^ neutrophils (**G**), and F4/80^+^ macrophages (**H**) isolated from the small bowel muscularis externa after cSTX-treatment (*n* = 5, 4). Values in each column are shown as mean ± SEM, and statistical analyses were carried out by ordinary one-way ANOVA (* *p* < 0.05, ** *p* < 0.01, and *** *p* < 0.001) comparing vehicle-treated mice versus indicated groups.

**Figure 3 ijms-22-06872-f003:**
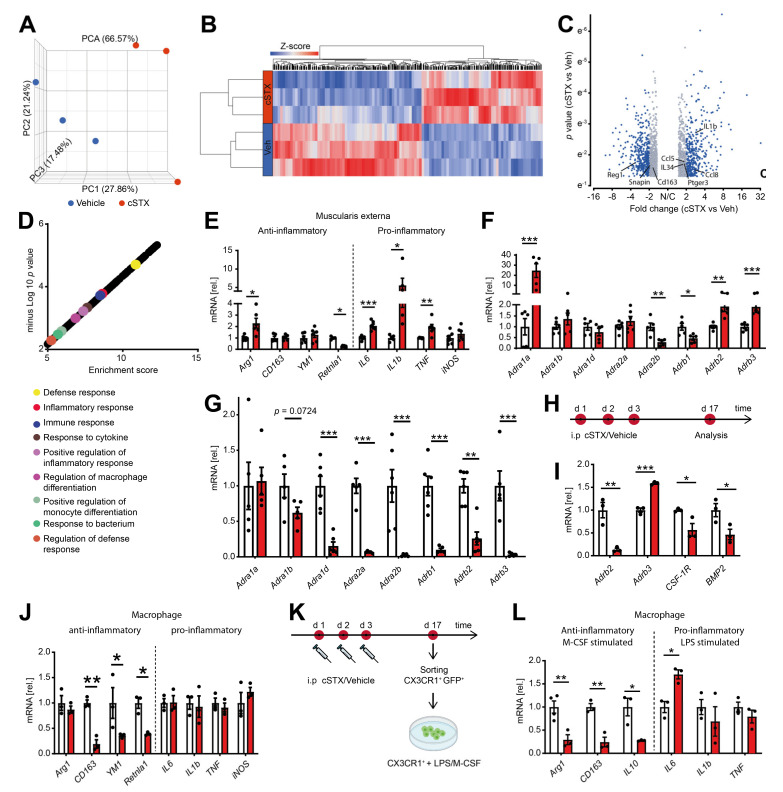
6-OHDA treatment modifies the basal immune response of the small intestinal muscularis and alters the inflammatory state of CX3CR1^+^ MMs. (**A**) Representative data of one of the two independent experiments (*n* = 3 per group) showing the principal component analysis (PCA) of a bulk RNA sequencing analysis from muscularis externa specimens of vehicle- (blue) and cSTX-treated animals (red). (**B**) Heat map analysis of differentially expressed genes showed the hierarchical clustering between vehicle- and cSTX-treated mice. (**C**) Volcano plot of differentially expressed genes, of which 775 were upregulated and 965 were downregulated in the cSTX-treated group as compared to the vehicle-treated group (*p* < 0.05, fold change ± 2). (**D**) Functional enrichment of immune-related GO terms. (**E**) qPCR analysis from vehicle- (white) and cSTX-treated mice (red) showing the levels of pro- and anti-inflammatory genes in the muscularis externa (*n* = 6). (**F**,**G**) qPCR analysis from vehicle- (white) and cSTX-treated mice (red) showing the altered gene expression of adrenergic receptors in the muscularis externa (*n* = 6) (**F**) and mucosa (*n* = 6) (**G**). Values in each column (**E**,**G**) are shown as mean ± SEM and statistical analysis was carried out by unpaired *t*-test (* *p* < 0.05, ** *p* < 0.01 and *** *p* < 0.001) comparing vehicle-treated mice versus indicated groups. (**H**) The experimental setting for sorting CX3CR1^+^ cells from intestinal muscularis externa, 17 days after vehicle or cSTX intraperitoneal (i.p) treatment. (**I**,**J**) mRNA expression (*n* = 3) of sorted macrophage receptors and cytokines from vehicle- (white) and cSTX-treated mice (red). (**K**) The experimental setting of an *ex vivo* LPS/M-CSF stimulation of sorted CX3CR1^+^ cells 17 days after vehicle or cSTX treatment. (**L**) qPCR analysis (*n* = 6) of sorted CX3CR1^+^ cells from vehicle- (white) and cSTX-treated mice (red) treated with LPS/M-CSF showing the levels of anti- and pro-inflammatory genes. Values in each column are shown as mean ± SEM, and statistical analysis was carried out by unpaired *t*-test (* *p* < 0.05, ** *p* < 0.01 and *** *p* < 0.001) comparing vehicle-sorted macrophages versus indicated groups (**I**,**J**,**L**).

**Figure 4 ijms-22-06872-f004:**
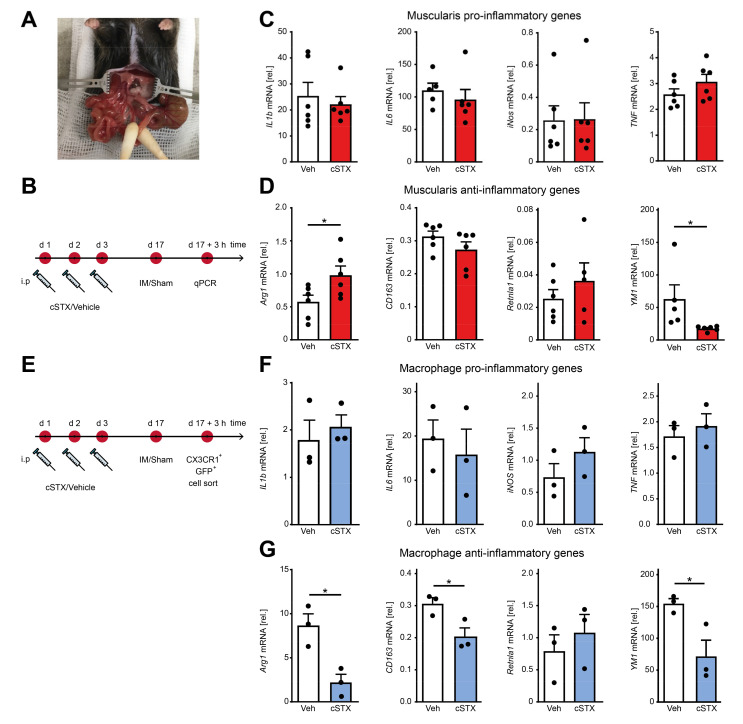
Effects of sympathetic denervation in the early phase of an acute intestinal postoperative ileus model. (**A**) Picture of an intestinal manipulation (IM) procedure to induce postoperative ileus as a model to study the role of SNS in acute bowel wall inflammation. (**B**) The experimental setting for the vehicle- (white) or cSTX-treatment (red) followed by IM. (**C**) qPCR analysis showing the mRNA expression of prototypic pro-inflammatory markers from the small intestinal muscularis samples of vehicle- (white) or cSTX-treated (red) mice 3 h after IM (*n* = 6). (**D**) qPCR analysis showing the mRNA expression of classical anti-inflammatory markers from the small-intestinal muscularis samples of vehicle- (white) or cSTX-treated (blue) mice 3 h after IM (*n* = 6). Values in each column are shown as mean ± SEM, and statistical analysis was carried out by unpaired *t*-test (* *p* < 0.05) (**C**,**D**) comparing vehicle-treated mice without IM versus indicated groups. (**E**) The experimental setting for sorting CX3CR1^+^ cells from intestinal muscularis externa of the vehicle- or cSTX-treated mice 3 h after IM. (**F**,**G**) qPCR analysis from sorted CX3CR1^+^ macrophages of vehicle- (white) or cSTX-treated (blue) mice 3 h after IM showing the mRNA expression of (**F**) pro-inflammatory and (**G**) anti-inflammatory markers (*n* = 3). Values in each column are shown as mean ± SEM, and statistical analysis was carried out by unpaired *t*-test (* *p* < 0.05) (**F**,**G**) comparing vehicle-sorted macrophages (baseline) versus indicated groups.

**Figure 5 ijms-22-06872-f005:**
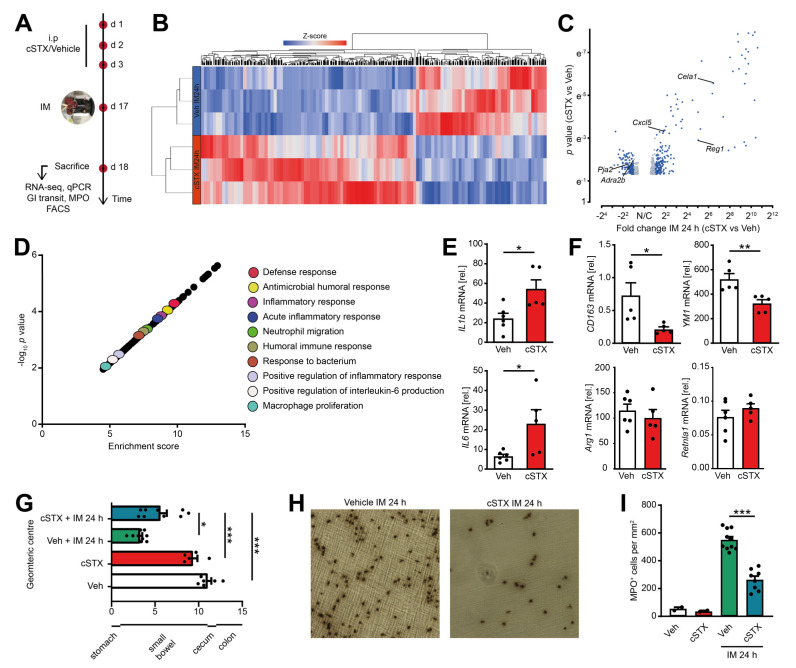
Effects of sympathetic neuronal depletion in the effector phase of POI. Mice were subjected to cSTX or vehicle (Veh) treatment before they underwent IM. (**A**) The experimental scheme to analyze the role of sympathetic innervation in POI. (**B**) Heat map analysis of differentially expressed genes showed the hierarchical clustering between vehicle- and cSTX-treated mice 24 h after IM. (**C**) Volcano plot of 456 differentially expressed genes between cSTX and vehicle-treated mice 24 h after IM, showing more than half of the genes upregulated in the cSTX-treated mice 24 h after IM. (**D**) Functional enrichment of immune-related GO terms. (**E**,**F**) qPCR analysis showing the mRNA expression of pro-inflammatory (**E**) and anti-inflammatory markers (**F**) from the small-intestinal muscularis samples of vehicle- (*n* = 6) (white) or cSTX-treated (*n* = 5) (red) mice 24 h after IM. Values in each column are shown as mean ± SEM, and statistical analysis was carried out by unpaired *t*-test (**p* < 0.05, ***p* < 0.01, and ****p* < 0.001) (**E**,**F**) comparing vehicle-treated mice without IM versus indicated groups. (**G**) GI transit time calculated 24 h after IM and shown as the geometric center of distribution of FITC dextran in the stomach (st), small intestine, cecum (c), and colon (*n* = 9). (**H**) Ileal muscularis whole mounts of vehicle- (left) and cSTX-treated (right) mice 24 h after IM showing myeloperoxidase-staining (MPO) for polymorphonuclear neutrophils (PMNs). (**I**) Quantification of MPO^+^ leukocytes in cSTX compared to vehicle-treated mice 24 h after IM. The graphs are plotted as mean ± SEM, and statistical analysis was carried out by two-way ANOVA (* *p* < 0.05, ** *p* < 0.01, and *** *p* < 0.001), (**G**,**I**) comparing vehicle-treated mice with indicated groups.

**Figure 6 ijms-22-06872-f006:**
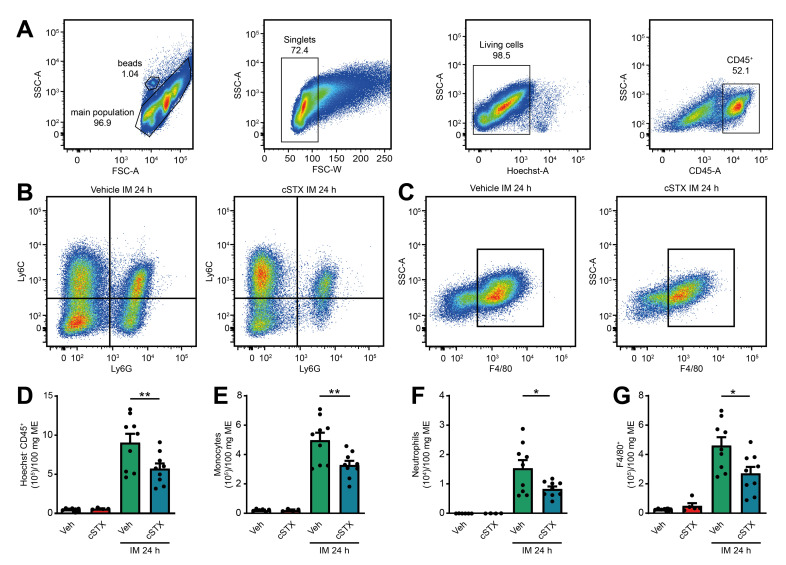
Effects of sympathetic neuronal depletion in the effector phase of POI. (**A**–**C**) Representative FACS images showing the gating strategy of the main population, singlets, and living CD45^+^ cells (**A**), the ratio of Ly6C^+^ Ly6G^-^ monocytes and Ly6C^+^ Ly6G^+^ neutrophils (**B**), monocyte-derived F4/80^+^ macrophages (**C**), pre-gated on Hoechst^-^ CD45^+^ of the vehicle- and cSTX-treated mice 24 h after IM. (**D**–**G**) Bar graphs showing the absolute numbers of total Hoechst^-^ CD45^+^ leukocytes (**D**), Ly6C^+^ Ly6G^-^ monocytes (**E**), Ly6C^+^ Ly6G^+^ neutrophils (**F**), and monocyte-derived F4/80^+^ macrophages (**G**) isolated from small bowel muscularis externa of vehicle- and cSTX-treated mice 24 h after IM (*n* = 9). The graphs are plotted as mean ± SEM, and statistical analysis was carried out by two-way ANOVA (* *p* < 0.05, ** *p* < 0.01).

**Figure 7 ijms-22-06872-f007:**
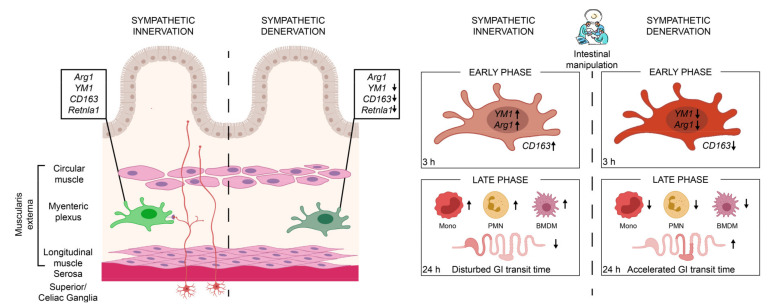
In the presence of an intact sympathetic innervation (red neurons), muscularis macrophages (light-green macrophage) express anti-inflammatory genes *Arg1, YM1, CD163*, and *Retnla1* that are reduced upon sympathetic denervation (dark-green macrophage). These muscularis macrophages are early responders upon surgical trauma (intestinal manipulation) of a noninfectious induced inflammation, clinically known as POI. Following the intestinal manipulation, these macrophages become activated and express high levels of alternative activation marker *Arg1* and *YM1* (light-red macrophage). Preoperative sympathetic denervation reduces these macrophage genes 3 h after intestinal manipulation resulting in a reduced expression of anti-inflammatory genes (dark-red macrophage). In the acute, clinically relevant effector phase of POI, the numbers of blood-derived monocytes (mono), neutrophils (PMN), and monocyte-derived F4/80^+^ macrophages (BMDM) are reduced in the muscularis which led to an accelerated transit time 24 h after intestinal manipulation and improved symptoms of POI.

## Data Availability

The data presented in this study are available on request from the corresponding author.

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
