# Peer review of "Sympathetic Denervation Alters the Inflammatory Response of Resident Muscularis Macrophages upon Surgical Trauma and Ameliorates Postoperative Ileus in Mice"

_ijms, 2021, doi:10.3390/ijms22136872_

Round 1

Reviewer 1 Report

COMMENTS FOR THE AUTHORS

In the present study, Mallesh et al., investigated the role of sympathetic nervous system (SNS) in the modulation of resident macrophage (MM) activation during postoperative ileus (POI), a transient motility disorder of the muscularis externa that occurs in response to intestinal manipulation. The authors compared three different sympathectomy (STX) models and identified a chemical approach as the most appropriate one. They showed that sympathetic depletion induces a reduction in the number of infiltrating leukocytes after intestinal manipulation with consequent improve of gastrointestinal transit. Taken together, the results of the present study suggest that the depletion of the sympathetic innervation affects the MMs immune state, reducing the cellular inflammation and, consequently, improving the postoperative outcome of POI.

The manuscript is interesting and it presents points of novelty.

I have some minor points:

_ Could the authors better discuss the hypothesized mechanisms through which sympathetic depletion regulates the MM immune state?

_ The authors should justify the choice of the range of concentrations used for the chemical denervation experiments and report the articles present in literature on which they are based.

_ Please check the statistical analysis used in figure 1E and 1F (see line 281, page 8). Did the authors use two-way ANOVA or T-test for the figure 1E and 1F?

_ Please explain the meaning of the abbreviations “SNS” line 16

_ Please replace the wording “TNF-a” with “TNF”

_ Line 278 page 8, please uniform the name of 6-OHDA

_ Line 334 page 13, “Intestinal manipulation” in lowercase

_ Please use the italics for the “ex vivo” and “in vitro”

Reviewer 2 Report

The article investigates the effect of sympathectomy on the inflammatory mechanisms in the gut. The investigation represents a good advance on the influence of sympathetic nerves on the intestinal inflammatory processes both in normal and in animals with experimental postoperative ileus. 

The work is performed in a very professional way and the results are all very credible and could be validated by other investigators.  The work is based on detailed analysis with modern methods of cellular physiopathology.

Minor points

When writing “defining the enteric (ENS), sympathetic (SNS), and parasympathetic nervous system (PNS) as the three anatomically distinct branches of the autonomic nervous system [1].” It makes reader thinking that the quoted article makes the distinctions of the three branches of the ANS. Langley in 1921 was the one that made this distinction clear.

Line 87: the term attenuation in “attenuation of TH+ cell fibers  in the intestine…” should perhaps replaced with ‘reduction’.

Line 94: “we applied the neurotoxin 6-OHDA intraperitoneally”, better we injected?

Line 95 “in ileal muscularis whole mounts” better ‘in whole mount preparations of small intestine’.

Line 500 where you write ” sympathetic innervation of the muscularis externa modulates MMs immune function”, the targeting of muscularis externa seems to me both incorrect and unnecessary. The sympathetic denervation surgical or chemical does not distinguish between the targets, muscle/myenteric plexus for motility - submucous plexus for secretion and perivascular nerves for blood flow. Remove the sentence “of the muscularis externa”.

The figures with TH immunoreactivity are too dark and small.
